# Seasonal patterns of bird and bat collision fatalities at wind turbines

**John D. Lloyd**[1], **Ryan Butryn**[1]*, **Schuyler Pearman-Gillman**[1,2], **Taber D. Allison**[1]

**1** Renewable Energy Wildlife Institute, Washington, DC, United States of America, **2** Apex Resource Management Solutions Ltd., Ottawa, Ontario, Canada

* rbutryn@rewi.org

## Abstract

Information on when birds and bats die from collisions with wind turbines can help refine efforts to minimize fatalities via curtailment of energy productions and can offer insight into the risk factors associated with collision fatalities. Using data pooled from 114 post-construction monitoring studies conducted at wind facilities across the United States, we described seasonal patterns of fatalities among birds and bats. Bat fatalities peaked in the fall. Silver-haired bat (*Lasionycteris noctivagans*), a long-distance migrant, and Mexican free-tailed bat (*Tadarida brasiliensis*) both showed maximum fatality counts later in the year–October and November, respectively–than any other bat species. The other common species in our sample–hoary bat (*Aeorestes cinereus*), Eastern red bat (*Lasiurus borealis*), and big brown bat (*Eptesicus fuscus*)–showed broadly overlapping peaks of fatality counts in August. Fatalities of silver-haired bat showed a smaller spring peak in some ecoregions; no other bat species exhibited this pattern. Seasonal patterns of bird fatalities varied among guilds. Woodland birds, many of which were long-distance migrants, showed two peaks in fatalities corresponding to spring and fall migration. Grassland birds and soaring birds, most of which were resident or short-distance migrants, did not exhibit strong seasonal peaks in fatalities. Species in these guilds tend to inhabit regions with extensive wind-energy development year-round, which may explain the more consistent numbers of fatalities that we observed. Our results highlight the value of pooling data to develop science-based solutions to reduce conflicts between wind-energy development and wildlife but also emphasize the need for more extensive data and standardization of post-construction monitoring to support more robust inferences regarding wind-wildlife interactions and collision risk.

## Introduction

Collision fatalities among birds and bats have been an incidental effect of wind energy since the first large-scale deployments of wind turbines. Several decades later, minimizing collision fatalities while maximizing energy production remains a key challenge in efforts to reconcile wildlife conservation with the rapid increase in wind energy that is needed to slow global warming [1, 2]. An important step in doing so is understanding why birds and bats collide with turbines in the first place. In temperate regions of the U.S. and Canada, a common and

**Data Availability Statement:** The data used in this study were obtained from the American Wind Wildlife Information Center (AWWIC). The AWWIC summarizes bird and bat data collected at 248 operating U.S. wind energy projects (more details

are found at: https://rewi.org/resources/awwic-bat-technical-report/). Tables S2 and S3 summarize the data available for this analysis. The data sourced from AWWIC were supplied to the authors through a data-sharing agreement. In order to request data used in this article, researchers will need to submit a detailed proposal to REWI at info@rewi.org or respond to a Renewable Energy Wildlife Research Fund request for proposals. AWWIC is a collaborative of independent data owners, managed by REWI, therefore data usage proposals will be evaluated and approved by all data owners.

**Funding:** The authors received no specific funding for this work.

**Competing interests:** The authors have declared that no competing interests exist.

fruitful approach in developing hypotheses about collision risk has been to examine the seasonal timing of fatalities (e.g., [3–8]).

Three general patterns have emerged from this body of research. First, birds, as a group, show two peaks in fatalities that correspond with spring (May) and autumn (September) migration, with the autumn peak in fatalities generally exceeding the spring peak in magnitude [6, 7]. Species- or family-specific patterns are difficult to resolve due to relatively small sample sizes, but in most cases seem to correspond roughly with the overall bimodal pattern [6]. This pattern highlights the spatial element of risk–where turbines are located relative to migratory routes may be important–and the risk associated with flight behavior during migration.

Second, bats, as a group, have a single, but lengthy, peak of fatalities from mid- to late summer until early autumn (mid-July to early September) that corresponds with both seasonal movements to wintering areas and mating periods. Bats do not show consistent evidence of a spring peak in fatalities, even though most of the bats killed in collisions with wind turbines undertake seasonal migrations (e.g., hoary bat [*Aeorestes cinereus*], Eastern red bat [*Lasiurus borealis*], and silver-haired bat [*Lasionycteris noctivagans*]) [3, 4, 7, 8]. Seasonal patterns of fatalities among bats have highlighted the potential importance of migratory and mating behavior as risk factors [9].

Finally, although some studies show no significant among-species variation in the timing of the main autumn peak in fatality counts for bats [7, 8], data from Baerwald and Barclay [4] indicate that silver-haired bat fatalities may peak later in the year than hoary bat: early September for the former vs. July or August for the latter. It is not clear why silver-haired bats may exhibit a pattern that differs from other migratory tree bats, but apparent differences in timing highlight the need to consider species-specific behaviors as an additional element of risk.

While previous research into the seasonal patterns in collision fatality rates has been useful for generating hypotheses about the causes of variation in collision risk, results have been based on small or geographically narrow samples. For example, one of the earliest descriptions of seasonal patterns in fatalities among bats was based on fatality-monitoring studies conducted at 19 wind-energy facilities across the U.S. and Canada [3]. More recently, Choi et al. [7] were able to examine fatality patterns among birds and bats at 44 operating facilities, but all of them were located in the northeastern United States. Similarly, Squires et al. [8] provided a highly resolved picture of species-specific variation in fatality counts of bats, but only for ten facilities in the southwestern portion of Ontario. To what extent results from local or regional studies reflect generally applicable patterns of risk, and thus can be extrapolated to describe seasonal patterns of collision risk in other areas, is uncertain but is important to understand given the rapid expansion of wind-energy generation across the U.S. [10]. In addition, fatality-timing patterns for some species, like Mexican free-tailed bat (*Tadarida brasiliensis*), remain largely undescribed because data from wind facilities within their range have not been widely available.

Our objectives in this analysis were to describe patterns in the seasonal timing of bird and bat fatalities at wind turbines and to offer insight into risk factors associated with collision fatalities. In doing so, we sought to support efforts to implement strategic curtailment of energy production by highlighting periods of time when collisions were most frequent and when curtailment potentially most efficient. In addition, we hoped to understand whether previously described patterns of collision timing were robust to a substantial increase in sample size and geographic extent. We did so by comparing our results with studies using smaller and more spatially restricted datasets.

Unlike previous efforts that have been limited to site-specific field research or the relatively small number of publicly available studies, we base our analysis on both public data and data contributed voluntarily by data owners–wind-facility operators or their delegates–to the

American Wind Wildlife Information Center (AWWIC). The American Wind Wildlife Information Center is a central database intended to make bird and bat collision fatality data at wind-energy facilities available for researchers while addressing the legal and reputational risk-management concerns of data owners. The database includes fatality records from a variety of bird and bat species collected across a broad geographic extent. Although it represents a self-selected sample due to the voluntary nature of data contributions, AWWIC is the most detailed, geographically extensive data set of its kind, containing data from 248 operating wind facilities in the United States that collectively represent nearly 30% of total installed capacity. As such, it offers opportunities to study broad patterns across the United States.

## Materials and methods

### Data source and preparation

Post-construction fatality monitoring (PCM) is conducted at industrial-scale wind-energy projects in the U.S. for the first one or two years of operation after construction, with additional years of monitoring occurring if required. Requirements for PCM vary by state and Federal permitting jurisdiction but have been a common practice since the Land-based Wind Energy Guidelines were published [11] (for an illustration and description of what constitutes a typical PCM, see [1]). In order to minimize potential bias introduced by variable PCM methods (e.g., different time intervals between searches), we limited our analysis to data in AWWIC for which a search schedule for the PCM was reported and for which exact dates on which carcasses were found were reported. The presence of a search schedule and exact dates of carcass discovery allowed us to precisely assign a date to each collision fatality that we analyzed and to associate daily search effort with daily carcass counts. The final dataset that we used included PCM from each year between 2009 and 2021.

In addition, we did not include carcasses found outside of a scheduled search because these discoveries did not have any associated measure of effort. To account for variable levels of effort on each search day all of our models (see below) included an offset term for the number of searches conducted per day, an effort to account for variable levels of effort on each search day. Thus, incidental carcass discoveries could not be included. If a PCM had search intervals of > 7 days, we also excluded records in which the estimated time since death was > 7 days. We did so in order to reduce the likelihood of including carcasses for which the date of death might be substantially mis-specified.

For bats, we included daily carcass counts for the five most frequently reported species (big brown bat [*Eptesicus fuscus*], Eastern red bat, hoary bat, Mexican free-tailed bat, and silver-haired bat) as well as for a group of the remaining bat species, which we refer to hereafter as "other bats". For a full list of the species included in the category of "other bats", see [12].

Sample sizes for most bird species were much smaller than those for bats so we did not conduct species-specific analyses for birds. Instead, as an initial exploration into variation in collision risk among bird species, we pooled species into three guilds that reflected broad similarities in life history and expected exposure to wind-energy facilities. A subset of 79 out of 306 total bird species in AWWIC was assigned to one of these guilds: woodland, grassland, or soaring (S1 Table). Guild assignments were made based on natural-history accounts in [13]. Due to small sample sizes, we did not include waterfowl, galliformes, shorebirds, waterbirds, or waders as guilds in this analysis. We also excluded several common generalist species that could not be assigned to one of our three guilds, including American Robin (*Turdus migratorius*), Rock Pigeon (*Columba livia*), European Starling (*Sturnus vulgaris*), and Brown-headed Cowbird (*Molothrus ater*) (for a full list of all bird species with fatality records in AWWIC, see

[14]). Finally, we also excluded any species reported as a collision fatality at only a single facility.

The woodland guild included songbirds, mostly passerines, that breed primarily in forested environments. Most were also long-distance migrants. The grassland guild consisted of grassland-nesting songbirds; most were short-distance migrants that winter and breed in grasslands in the United States. The soaring guild included mostly raptorial species with a variety of habitat preferences and migration strategies that were united in their tendency to use soaring flight and to hunt on the wing.

## Model development and selection

The response variable for this analysis was the daily count of carcass discoveries made each day a search was conducted. The explanatory variable of primary interest was day of year. We also considered species (bats) or guilds (birds) and level II North American ecoregions (https://www.epa.gov/eco-research/ecoregions-north-america) (Fig 1) as potential covariates affecting the relationship between carcass discoveries and time of year. We included species or guilds as a factor in some models because we anticipated, based on previous research, that species or guilds would show different seasonal patterns of fatalities.

We included ecoregion both to account for differences in species composition of fatalities across our study and because we expected that differences in climate among ecoregions might contribute to differences in seasonal activity of animals and thus patterns of carcass counts. To maintain anonymity of facilities reporting fatalities to AWWIC, we did not include in this analysis data from any ecoregion that had <5 facilities reporting results of PCM. The one exception to this rule was for the Texas-Louisiana Coastal Plain (9.5) ecoregion and the Tamaulipas-Texas Semiarid Plain (9.6), each of which had <5 facilities represented but which we believed could be combined to form a larger region that we refer to as Southern Texas Plains for this analysis. Doing so also allowed us to avoid discarding potentially useful data on species, like Mexican free-tailed bat, that otherwise are sparsely represented in AWWIC.

We chose to model seasonal patterns of bird and bat collisions with wind turbines using generalized additive mixed models (GAMMs; [15]). These models allow greater flexibility for non-linear distributions by using smoothing functions that closely follow the shape of the data and are not constrained by the assumption of linearity. The models assume correlation in time-series data points rather than considering each observation independently, which is well-suited to the daily counts of carcasses. GAMMs also allow for the inclusion of random effects, which may help account for unmeasured, idiosyncratic effects of site and year in which data were collected.

We used GAMMs first to describe seasonal variation in counts of all carcasses, both birds and bats. This served as a null model, including only day of year as a continuous smooth term, and ignoring all of the putatively important risk factors that we included in subsequent models. We considered two additional classes of models. First, we fit models that included factors for ecoregion, taxon (birds v. bats), and the interaction between ecoregion and taxon. Results of these models allowed us to examine differences between birds and bats in the seasonal timing of fatalities and to determine whether seasonal patterns varied among ecoregions. Second, we fit separate models for bats and birds, and again included day of year as a continuous smooth term and considered factors for ecoregion, species (or guild, in the case of birds), and the interaction between ecoregion and species or guild. This last class of models allowed us to investigate whether individual species or groups of species exhibited seasonal or geographical patterns of collision risk.

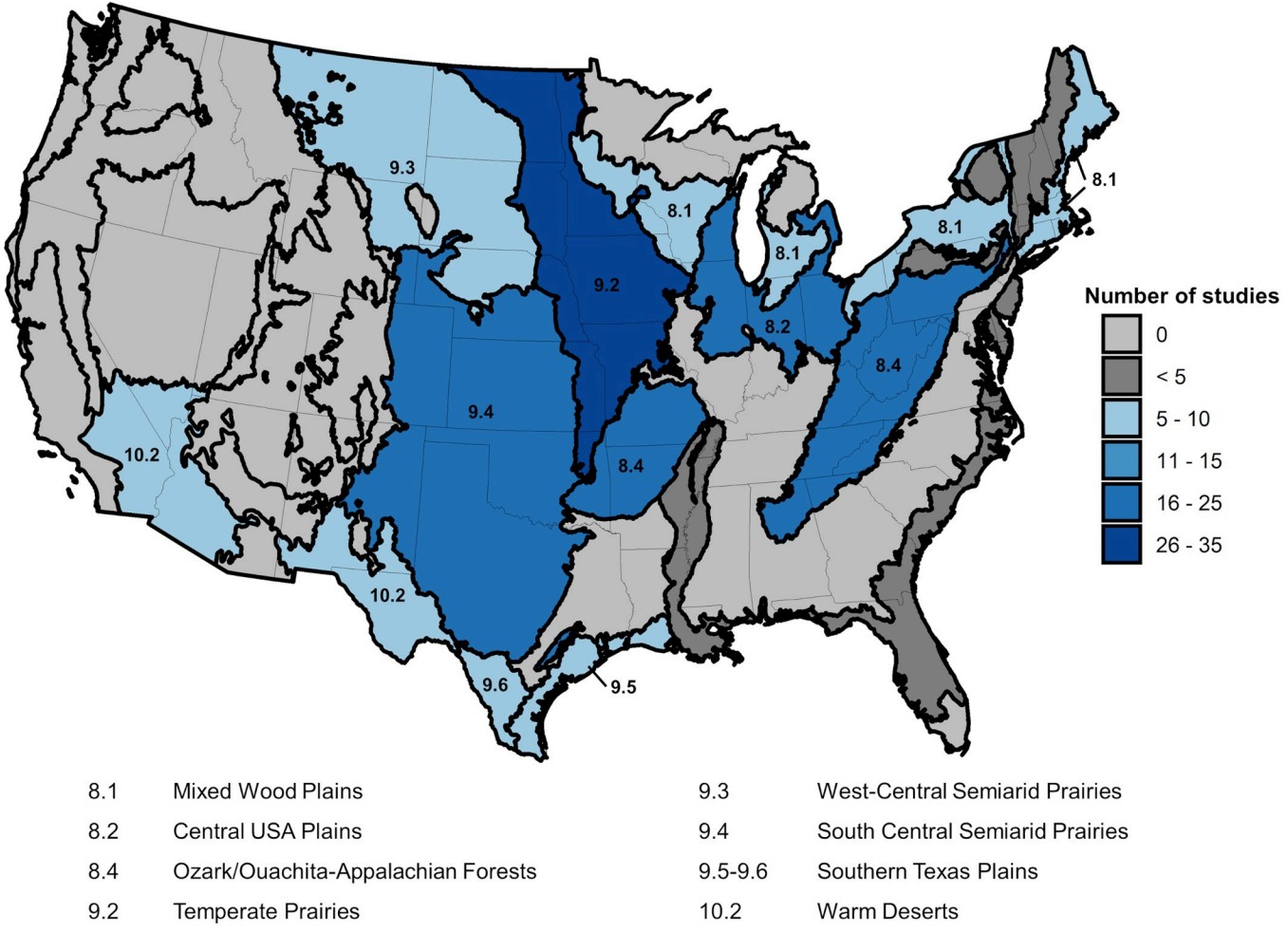

**Fig 1. Map of North American Level II ecoregions and number of post-construction fatality monitoring studies meeting analysis criteria in the conterminous USA.**

| | | | |
|---|---|---|---|
| 8.1 | Mixed Wood Plains | 9.3 | West-Central Semiarid Prairies |
| 8.2 | Central USA Plains | 9.4 | South Central Semiarid Prairies |
| 8.4 | Ozark/Ouachita-Appalachian Forests | 9.5-9.6 | Southern Texas Plains |
| 9.2 | Temperate Prairies | 10.2 | Warm Deserts |

All analyses were conducted in R (version 4.1.1; [16]) using the mgcv package (version 1.8–38; [15]). Due to the cyclical nature of day of year data (i.e., day 1 comes after day 365), all day-of-year smooth terms were fit with a cyclic cubic regression spline (bs = "cc"). Random effect terms were fit with random effect splines (bs = "re"). We used the "by" argument in the mgcv package to fit smooth trends to each supplied factor level of species or guild, ecoregion, and the combined interaction between species or guild and ecoregion.

We tested the carcass count data for overdispersion and assessed whether GAMMs that assumed either a Poisson or negative binomial distribution better fit the data. Due to variable levels of effort on each search day, we included an offset term for the number of searches conducted each day. The carcass counts could be influenced by the site they were collected at either due to methodology (e.g., survey protocols or searcher efficiency) [17] or by landscape characteristics (e.g., amount of habitat or terrain features) [18], so we also tested whether support for a model improved if we included site as a random effect. The year in which a study took place could also influence collision timing, for example due to weather patterns that changed seasonal patterns of movement or activity, and we therefore tested year of study as both a fixed and random effect in our GAMMs.

We used Akaike's Information Criterion (AIC) scores to rank competing parameterizations of each model class. To assess how robust our models were, we also ran the top model from each model class on a randomly selected validation dataset that represented 50% of the data (model selection was based on analyses conducted using the full dataset). Model results were considered robust and retained in the analysis only if the smoothing function term had a P-value <0.1 for both the full model and the validation model and if the normalized root mean square error (RMSE) of the validation model was within the range of the observed values (i.e., RMSE <1). We considered models that did not meet these criteria as representing potentially spurious relationships and we did not consider them further or include them in discussion of results.

## Results

After applying the data selection filters to the studies in the AWWIC PCM dataset, there were 114 studies available for analysis. The lack of a search schedule was the most common reason for censoring data (n = 198 studies). The remaining studies were excluded because of incomplete incident data or occurrence in an ecoregion with fewer than five facilities reporting PCM results. The included studies accounted for 14,721 search days and 312,510 total searches. In total, 10,291 bat carcasses and 3,789 bird carcasses with exact find dates were available for this analysis (S2 and S3 Tables and for total carcass counts by species, see [12, 14]). Eight ecoregions contained five or more studies, although not all species or species groups were represented in each ecoregion. Daily carcass count data were found to be over-dispersed and best fit with a negative binomial distribution. For all analyses, GAMMs that used both site and year as random effects had the lowest AIC scores (S4–S7 Tables), implying that site location and annual conditions may influence the timing of fatalities.

Analysis of the null model and its smooth function for day of the year indicated a primary peak of carcasses in mid- to late-August and a smaller, secondary peak in early May (Fig 2 and upper panel). This model had an overall $R^2$ = 0.33, explained 41.3% of deviance, and the validation split had an overall normalized RMSE of 0.056. Search effort varied throughout the year but did not explain seasonal peaks in bird and bat fatalities (Fig 2 and lower panel). The distribution of search effort throughout the year likely reflected limited winter accessibility in cold climates, reduced sampling during periods of low collision risk, or variable survey protocols among studies. Our models accounted for seasonal variation in search effort by including effort as an offset, although we avoided making predictions from model estimates for days with few or no searches (S1 and S2 Figs).

As expected, fitting separate curves for birds and bats improved model performance; including ecoregion as a variable was also supported by the data (S5 Table). The top performing model that estimated parameters for birds and bats separately had an overall $R^2$ = 0.38, explained 47.7% of deviance, and had an overall normalized RMSE of 0.041. Ecoregional models for all bats met our criteria for reliability, but three ecoregional models of bird fatalities were dropped due to high P-values for the full model or the validation model (S8 Table). For bats as a group, collisions peaked in late summer in all ecoregions considered (Fig 3 and Table 1). Seasonal patterns for birds were inconsistent across ecoregions (Fig 3 and Table 1).

Our final class of models, in which we fit separate models of fatality timing for bird guilds and bat species, offered further insight. Analyses of bat-carcass data suggested that an interaction of bat species and ecoregion provided the best model (S6 and S9 Tables). This model had an overall $R^2$ = 0.31, explained 57% of deviance, and had a normalized RMSE of 0.018. Hoary bat, Eastern red bat, and big brown bat showed a coincident pattern of maximum fatalities that occurred in mid-August (Fig 4). The peak for silver-haired bats consistently occurred later, in

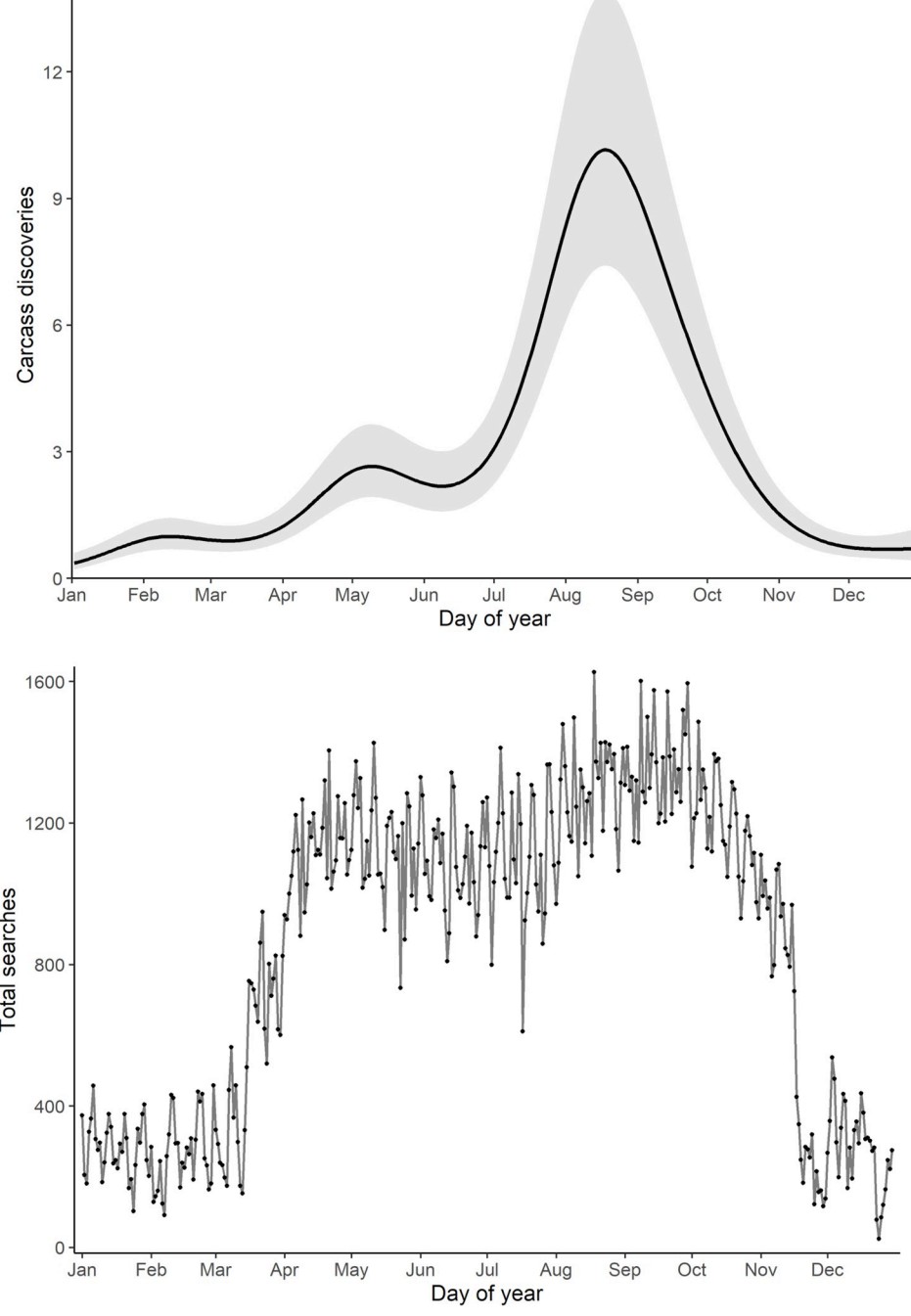

**Fig 2. Daily combined bird and bat carcass discovery predictions from generalized additive mixed model.** In the top panel, carcass discoveries are per 100 searches and shaded area is the 95% confidence band of the predicted counts. Bottom panel displays the total number of searches summed across all available studies for each day of the year.

mid- to late September (Fig 4). Although fatalities of Mexican free-tailed bat were uncommon in most of the regions that we considered, counts of this species seem to peak very late in the year (late October and early November) (Fig 4). Silver-haired bats, unlike any of the other bat species, showed a smaller, secondary peak in fatalities during spring in some of the ecoregions

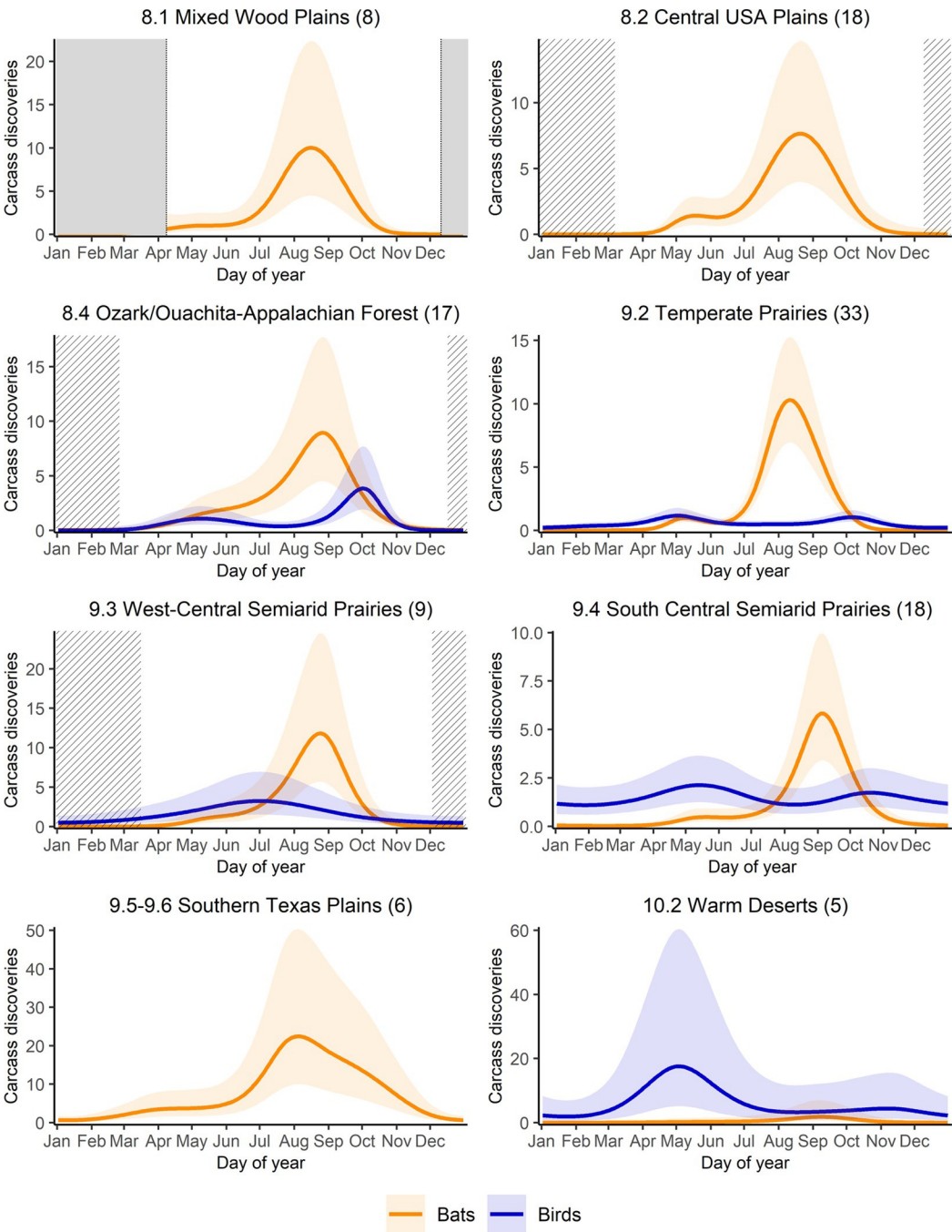

**Fig 3. Daily carcass discovery predictions from generalized additive mixed model for all bats and all birds by North American Level II ecoregion.** Carcass discoveries are per 100 searches and y-axis scale may vary among regions. Center line shows the predicted mean and shaded regions show the 95% confidence intervals. Solid grey sections indicate extended periods of no data. Striped sections indicate periods of sparse data. Only significant results are plotted.

we considered (Fig 4). Finally, at least among the migratory tree bats, fatalities peaked earlier in more northerly ecoregions (Table 1).

From analyses of bird carcasses only, the best-supported model included an interaction of guild and ecoregion (S7 Table). This model, however, had a low $R^2$ (0.14) and deviance

**Table 1. Peak collision dates calculated from GAMM models for each species or guild in each ecoregion.**

| | 8.1 Mixed Wood Plains | 8.2 Central USA Plains | 8.4 Ozark/ Appalachian Forests | 9.2 Temperate Prairies | 9.3 West-Central Semiarid Prairies | 9.4 South Central Semiarid Prairies | 9.5–9.6 Southern Texas Plains | 10.2 Warm Deserts |
|---|---|---|---|---|---|---|---|---|
| **Bats** | | | | | | | | |
| All bats | 16-Aug | 20-Aug | 27-Aug | 11-Aug | 24-Aug | 5-Sep | 5-Aug | 8-Sep |
| Hoary bat | 10-Aug | 5-Aug | 23-Aug | 12-Aug | 17-Aug | 4-Sep | — | — |
| Eastern red bat | 5-Aug | 16-Aug | 21-Aug | 6-Aug | 21-Aug | 28-Aug | — | — |
| Silver-haired bat fall | 14-Sep | 22-Sep | 17-Sep | 14-Sep | 5-Sep | — | — | — |
| Silver-haired bat spring | 30-May | 21-May | 2-Jun | 30-Apr | 18-May | — | — | — |
| Big brown bat | —[a] | 10-Aug | 2-Aug | 11-Aug | — | — | — | — |
| Mex. free-tailed bat fall | NP[b] | NP | NP | NP | NP | 12-Sep | 28-Sep | — |
| Mex. free-tailed bat spring | NP | NP | NP | NP | NP | 22-May | NA | — |
| Other bats | 30-Aug | — | 22-Aug | 23-Aug | NP | 2-Aug | 30-Jul | — |
| **Birds** | | | | | | | | |
| All birds fall | — | — | 2-Oct | 6-Oct | 2-Jul | 20-Oct | — | NA |
| All birds spring | — | — | 10-May | 2-May | NA | 14-May | — | 3-May |
| Grassland | — | — | — | 21-May | 20-Jun | 6-May | — | — |
| Soaring | — | — | — | — | — | — | — | — |
| Woodland fall | — | 27-Oct | 30-Sep | — | — | 26-Sep | 8-Oct | — |
| Woodland spring | — | NA[c] | 26-May | — | — | 15-May | 27-Apr | — |

[a]"—" = Model not significant

[b]NP = Species not present

[c]NA = No discernable peak

explained (38.2%). Bird guilds differed substantially in the seasonal timing of fatalities (Fig 5). Ecoregional differences, although identified as important by our model-selection criteria, are less pronounced and not readily interpretable (Fig 6). Indeed, most of the ecoregional guild models did not meet our criteria for reliability (S10 Table).

## Discussion

Based on data collected across the U.S. over 13 years (2009–2021), our findings replicate existing descriptions of temporal patterns of collision fatalities at wind turbines and reveal several new insights. We conclude that most of what is known about seasonal variation in collisions among birds and bats at wind facilities in the U.S. holds true even as sample size increases by an order of magnitude, supporting the use of these patterns in the inductive development of hypotheses concerning collision risk factors.

Pooling PCM data in AWWIC proved especially useful in allowing a finer-grained analysis of bird collisions. In particular, we found distinct patterns in the seasonal timing of collision fatalities among bird guilds, which has not been described previously. Seasonal peaks in fatality counts were most evident for birds that we classified into the woodland guild. This group of species, nearly all of which are long-distance migrants between breeding locales in the Nearctic and wintering areas in the Neotropics, showed spring and fall peaks in fatality counts that correspond to the typical timing of spring and autumn migration. Grassland and soaring guilds, which included few long-distance migrants, did not exhibit strong seasonal peaks in carcass

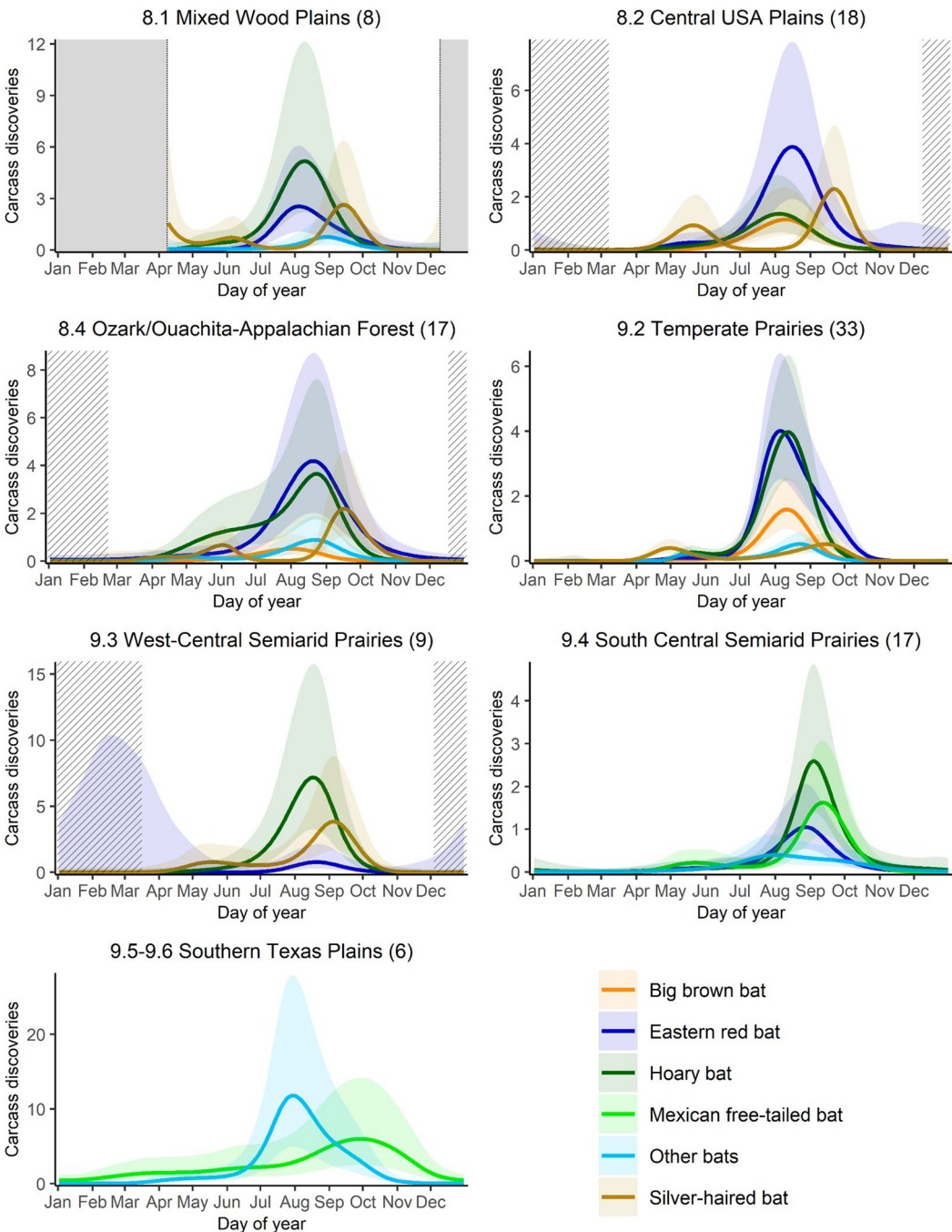

**Fig 4. Daily carcass discovery predictions from generalized additive mixed model for bat species by North American Level II ecoregion.** Carcass discoveries are per 100 searches and y-axis scale may vary among regions. Center line shows the predicted mean and shaded regions show the 95% confidence intervals. Solid grey sections indicate extended periods of no data. Striped sections indicate periods of sparse data. Only significant results are plotted.

discoveries. The importance of migration as a risk factor for birds has been noted previously [6, 7] but the differences that we observed among bird guilds suggests that the association between bird migration and collision fatalities also reflects where wind-energy facilities are constructed and operated. With most facilities in the U.S. situated in open landscapes of the

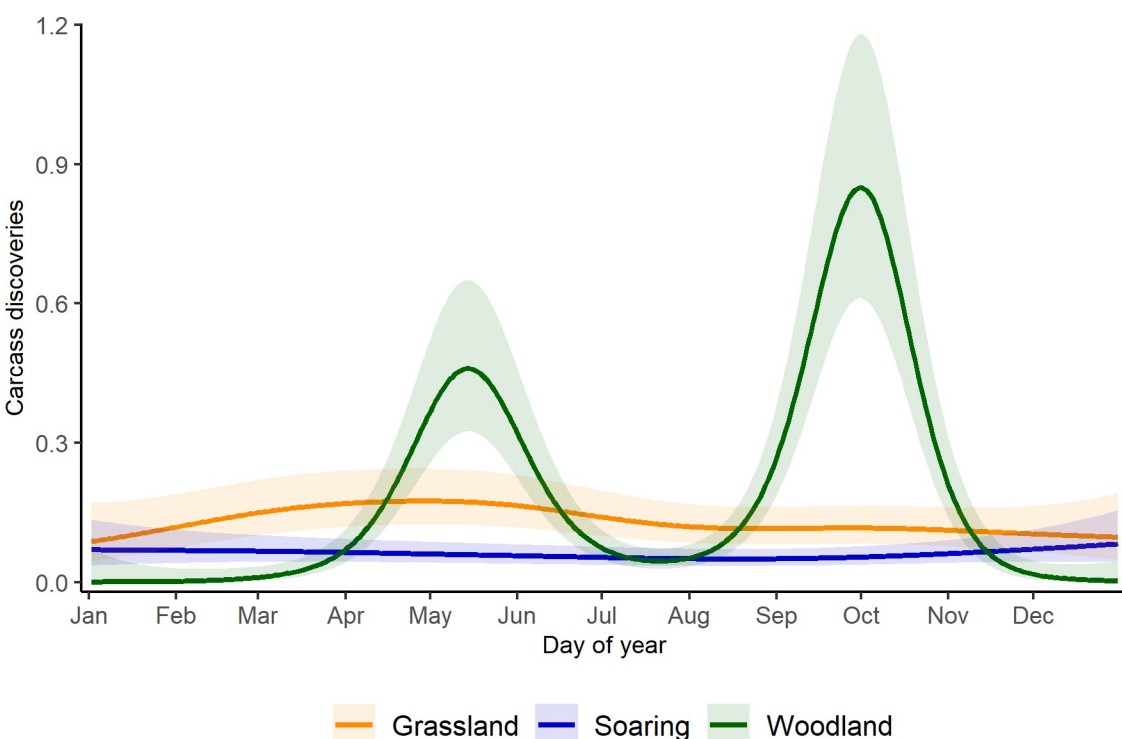

**Fig 5. Daily carcass discovery predictions from generalized additive mixed model for bird guilds in conterminous USA.** Carcass discoveries are per 100 searches, center line shows the predicted mean and shaded regions show the 95% confidence intervals.

central and western states, long-distance migrants that breed in forested areas will only encounter turbines while moving between breeding and wintering areas. In contrast, many species in the grassland and soaring guilds are present year-round in regions of the U.S. that host significant numbers of wind turbines [19]. As such, species in these guilds may experience relatively consistent exposure to wind turbines and, as a consequence, relatively consistent levels of collision risk among seasons of the year. The hypothesis that exposure to turbines drives interspecific variation in collision fatalities of birds could be tested by examining the relationship between collision fatalities and the amount of time different species spend in the vicinity of operating turbines, for example by combining data on the location of turbines in the U.S. [15], collision fatalities in AWWIC, and intra-annual distribution models generated by eBird (eBird Status and Trends products; ebird.org/science/status-and-trends).

The sample size for bat fatalities recorded in AWWIC was much larger than that for birds and patterns in collision timing generally matched those reported by previous analyses of smaller data sets [3, 4, 7, 8], with fatalities peaking in late summer and early autumn. As has been observed in other studies (e.g., [4]), we also found that collision fatalities of silver-haired bats peaked later in the year than for both the other migratory tree bats (hoary bat and Eastern red bat) and the one cave-roosting species in our sample, big brown bat. However, the geographically extensive data in AWWIC allowed us to compare for the first time the timing of fatalities for these more widespread species with Mexican free-tailed bats, carcasses of which appeared in large numbers only in the southernmost regions of our study area. Mexican free-tailed bats exhibited the latest peak in fatality counts, with most carcasses discovered during late October and early November. This fits with previous observations that the intensity of autumn migration in the U.S. peaks for this species during October and November [20, 21].

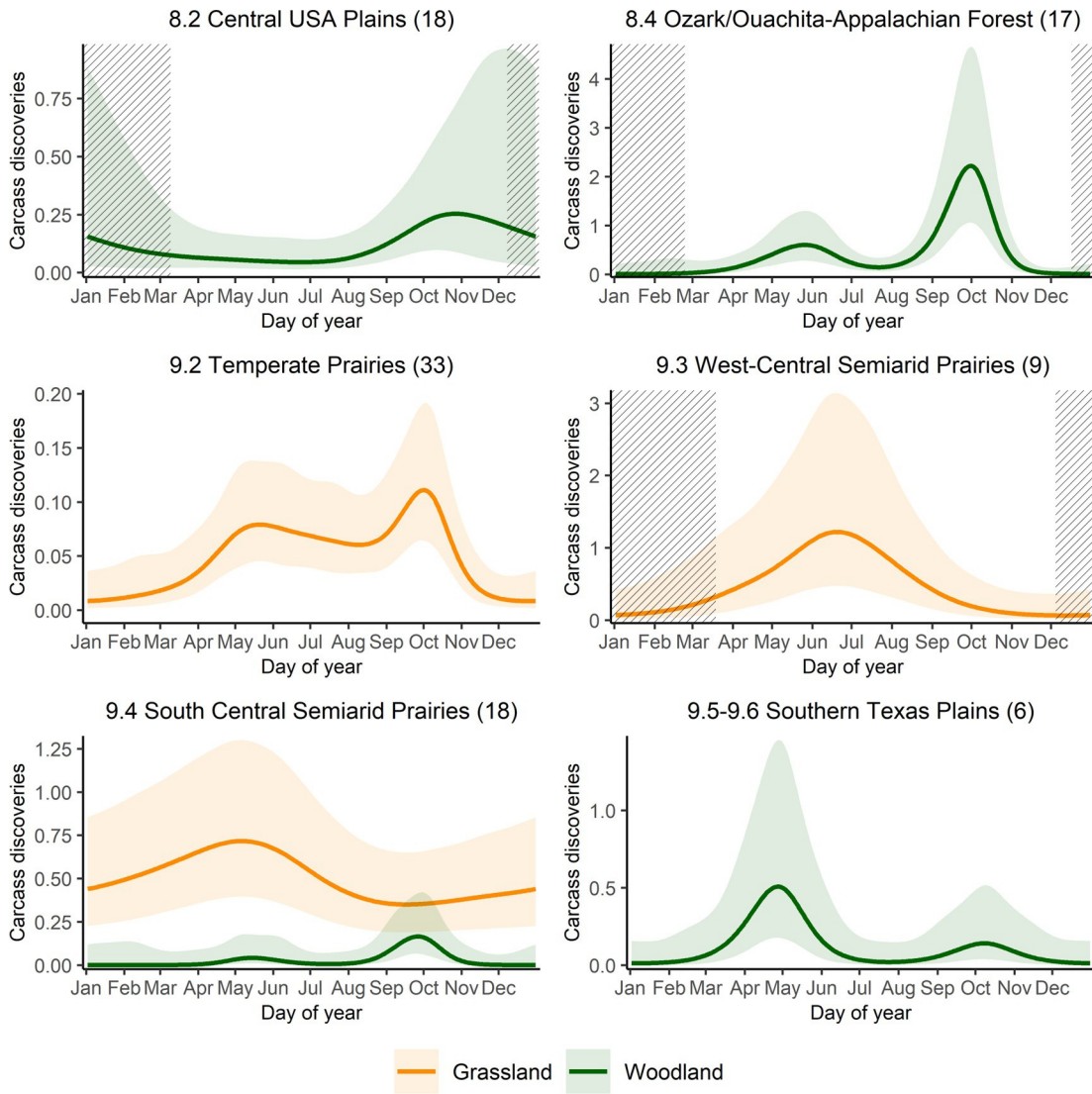

**Fig 6. Daily carcass discovery predictions from generalized additive mixed model for bird guilds by North American Level II ecoregion.** Carcass discoveries are per 100 searches and y-axis scale may vary among regions. Center line shows the predicted mean and shaded regions show the 95% confidence intervals. Striped sections indicate periods of sparse data. Only significant results are plotted.

Mexican free-tailed bats have begun overwintering in larger numbers in Texas, rather than continuing to migrate further south into Mexico, which may further increase the duration of time during which this species is at risk of collisions [22].

Using data from a large geographic area also allowed us to examine how temporal patterns in the number of collisions varied across latitudes, a question that cannot be addressed by site- or region-specific studies. Among the migratory tree bats, species that occurred across multiple ecoregions tended to show earlier fatality peaks in more northerly ecoregions. For example, when we compared the estimated peak date of fatalities for hoary bat between adjacent ecoregions, the more northerly ecoregion always showed an earlier peak date. The peak date of hoary bat fatalities in the West-Central Semiarid Prairies was 17 August versus 4 September in the South Central Semiarid Prairies; 12 August in the Temperate Prairies versus 23 August in

the Ozark/Ouachita-Appalachian Forests; and 10 August in the Mixed Wood Plains versus 23 August in the Ozark/Ouachita-Appalachian Forests. The same patterns were evident for both Eastern red bat and silver-haired bat. We assume that this latitudinal gradient in the timing of maximum fatality counts reflects the gradual southward movement of a broad migratory front of bats, perhaps triggered by favorable weather conditions that encouraged large numbers of individuals to initiate large-scale movements [4, 9, 23].

Analyzing data from across the U.S. also offered new insight into the potential to apply region-specific curtailment practices. When bat fatalities are considered at a national scale, regional and interspecific differences in the timing of fatalities result in a fairly long period of time during which carcass discoveries are elevated. For example, discoveries of carcasses of hoary bat, Eastern red bat, and big brown bat begin to increase in frequency by mid- to late-June in northern regions, whereas the number of Mexican free-tailed bat carcasses found in south Texas only declines towards baseline levels beginning in mid-November. This suggests a nearly 5-months-long period during which bat fatalities at wind facilities in the U.S. are elevated. Region-specific curtailment requirements would shorten the duration of curtailment at any individual facility, which at present is typically instituted during some period of time from July–October [24]. However, complicating translation of our results into practice is the finding that the timing of carcass discoveries varied by location and year. In other words, our best-supported model indicated that the seasonal timing of fatalities varied from year-to-year and among sites, which limits our ability to predict future peaks in fatalities at any given site. Optimizing curtailment to minimize bat fatalities and loss of electricity production at any individual facility may thus require location-specific data collected over multiple years. Absent ongoing PCM at many locations across the U.S., longer curtailment periods will be required to account for uncertainties in when fatalities will peak at any given site in any given year.

Our analysis also highlighted some of the challenges and opportunities associated with mining the large number of fatality records contained in AWWIC. Generalized additive mixed-models proved useful in visualizing complex, multimodal distributions and provided a framework for testing explanatory variables while controlling for variation in search effort. However, we also discovered that fewer than one-third of available datasets met our standards for inclusion (114 of 370 studies), largely because they lacked associated search schedules. We explored options that would have allowed us to include more of the available data, for example approximating search dates and back-filling with zero finds, but, in most cases, we were unable to reliably assign a search date to all turbines included in a PCM. Going forward, all new studies contributed to AWWIC will be required to include search schedules and exact dates of carcass discovery, which will ensure that future efforts to model collision timing will not be restricted for similar reasons.

The consequences of censoring a large number of records appeared relatively minor in this analysis. Our model validation indicated that results for bat models are robust for most regions and species and not sensitive to the amount of data. Data availability was limiting for bird guild models, although given the low numbers of bird carcasses found compared to bats [12, 14], it is unlikely that all results would be significant even if all AWWIC studies met the requirements. Gains in inferential ability regarding geographic variation in the timing of bird fatalities might be realized by using a larger spatial unit than the level II ecoregion (e.g., avifaunal biomes used in [6]) because adopting a coarser approach to forming groups for analysis would yield larger sample sizes while maintaining the ability to explore differences associated with geography, climate, and species composition.

Including the daily number of searches as an offset term in the models served as a proxy for the variability in daily search effort. Previous efforts to describe seasonal variation in the timing of collision fatalities have not been able to account for search effort and have thus been

subject to the potential bias of mistaking increased search effort for increased numbers of collision fatalities. We would have preferred to use the daily area searched so that our measure of effort accounted for different plot sizes, but most of the studies in AWWIC use multiple plot sizes and rarely were data reported in a format that allowed us to determine the plot size of every search. This made it impossible to calculate daily area searched in all cases. Although it is unclear whether representing search effort with daily area searched would improve the performance of the models substantially, doing so would be a more accurate measure of effort given that many recent studies have adopted a variety of search strategies (e.g., surveying only on roads and turbine pads vs. carrying out complete, area-based searches around turbines).

Data used in this analysis were drawn from PCM conducted at facilities with variable curtailment regimes. Some imposed curtailment, whereas others did not. However, we do not believe that variation in curtailment regime was likely to have introduced substantial bias into our results. First, any effect of curtailment should be negligible for birds. In the U.S., curtailment is typically imposed during the evening to reduce bat fatalities. The largely diurnal habits of the birds considered in this analysis and the tendency of nocturnal migrants to fly well above the rotor-swept area should mean that curtailment as currently practiced will have negligible effects on bird fatalities. Some empirical evidence supports this assumption [25].

Curtailment is generally effective in reducing bat fatalities, however [26], and thus may have affected the number of carcasses recovered during PCM. Nonetheless, we think that variation in curtailment regimes is unlikely to affect our results or conclusions. Curtailment in the U.S. is usually imposed during a specific window of time when bat fatalities are expected to be greatest, typically July–October [24]. If effective, curtailment should depress seasonal peaks in fatality counts but will not introduce spurious peaks in carcass counts. This means that curtailment would most likely render our tests for significant seasonal peaks conservative, making it more difficult to discern statistically significant peaks in carcass counts. It should not, however, create artificial patterns of seasonality.

Our step-wise, hierarchical approach to modeling the seasonal timing of collision fatalities hints at the potential for further refinement of the patterns we found. Our three classes of models represented increasingly finer taxonomic and spatial divisions in patterns of fatality timing, and at each stage of modeling we found support for including more complex models. Yet, despite access to a much larger sample than has previously been analyzed, in the end we were unable to comprehensively describe regional and species-specific differences in the timing of fatalities even though model-selection criteria supported such models. For example, most of the ecoregion-by-guild models for bird fatalities failed to produce consistent results when applied to the validation dataset. In part, our inability to generate robust inferences about subsets of the data arose because absolute sample sizes remain small, especially for birds; most bird species included in AWWIC are represented by <100 carcasses [14]. Small sample sizes also arise because the geographic distribution of facilities and studies included in AWWIC is uneven, with some ecoregions drawing on data from only five studies.

Although our results demonstrate the potential value of pooling fatality data from post-construction monitoring analyses, they also highlight that small sample sizes continue to limit our ability to describe fatality patterns at fine spatial and taxonomic resolutions. Continuing to expand the database with new studies, and requiring that all contributed studies include necessary metadata about search schedules, might allow future studies to examine region- and taxa-specific patterns of fatality timing in greater detail.

## Supporting information

**S1 Fig. Total daily searches per ecoregion.**
(TIF)

**S2 Fig. Number of daily turbine searches per ecoregion.**
(TIF)

**S1 Table. Bird species included in the Grassland, Soaring, and Woodland guilds.**
(DOCX)

**S2 Table. Summary of data available for model development for all-bat and all-bird models.**
(DOCX)

**S3 Table. Summary of data available for model development in bat species and bird guild models.**
(DOCX)

**S4 Table. Model selection results for base models (without grouping data by species).**
(DOCX)

**S5 Table. Model selection results for all-bat/all-bird models.**
(DOCX)

**S6 Table. Model selection results for bat species models.**
(DOCX)

**S7 Table. Model selection results for bird guild models.**
(DOCX)

**S8 Table. Significance of predictor variables for full dataset models (All) and the 50/50 split validation dataset (Split) for the all-bat/all-bird models.**
(DOCX)

**S9 Table. Significance of predictor variables for full dataset models (All) and the 50/50 split validation dataset (Split) for bat species models.**
(DOCX)

**S10 Table. Significance of predictor variables for full dataset models (All) and the 50/50 split validation dataset (Split) for bird guild models.**
(DOCX)

## Acknowledgments

We thank all of the data contributors to AWWIC for providing data and allowing us to include it in this analysis. Shilo Felton provided a useful critique of an earlier draft of this manuscript.

## Author Contributions

**Conceptualization:** John D. Lloyd, Ryan Butryn, Taber D. Allison.

**Data curation:** Ryan Butryn, Schuyler Pearman-Gillman.

**Formal analysis:** Schuyler Pearman-Gillman.

**Investigation:** John D. Lloyd, Ryan Butryn, Schuyler Pearman-Gillman.

**Methodology:** John D. Lloyd, Ryan Butryn, Schuyler Pearman-Gillman, Taber D. Allison.

**Project administration:** John D. Lloyd, Ryan Butryn, Taber D. Allison.

**Resources:** Ryan Butryn, Schuyler Pearman-Gillman.

**Software:** Ryan Butryn, Schuyler Pearman-Gillman.

**Supervision:** John D. Lloyd, Ryan Butryn, Taber D. Allison.

**Validation:** John D. Lloyd, Ryan Butryn, Schuyler Pearman-Gillman.

**Visualization:** Schuyler Pearman-Gillman.

**Writing – original draft:** John D. Lloyd.

**Writing – review & editing:** John D. Lloyd, Ryan Butryn, Schuyler Pearman-Gillman, Taber D. Allison.

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
