## [Decision Letter · Decision Letter 0]

30 Jan 2023

PONE-D-22-30951Seasonal patterns of bird and bat collision fatalities at wind turbinesPLOS ONE

Dear Dr. Lloyd,

Thank you for submitting your manuscript to PLOS ONE. After careful consideration, we feel that it has merit but does not fully meet PLOS ONE’s publication criteria as it currently stands. Therefore, we invite you to submit a revised version of the manuscript that addresses the points raised during the review process. Four experts have reviewed your manuscript. Three of the four offer minor suggestions, but Reviewer 2 raised a few significant concerns that need to be addressed. Please address all reviewer comments in your revisions.

We look forward to receiving your revised manuscript.

Kind regards,

William David Halliday, Ph.D.

Academic Editor

PLOS ONE

Journal Requirements:

4. We note that Figure 1 in your submission contain map images which may be copyrighted. All PLOS content is published under the Creative Commons Attribution License (CC BY 4.0), which means that the manuscript, images, and Supporting Information files will be freely available online, and any third party is permitted to access, download, copy, distribute, and use these materials in any way, even commercially, with proper attribution. For these reasons, we cannot publish previously copyrighted maps or satellite images created using proprietary data, such as Google software (Google Maps, Street View, and Earth). For more information, see our copyright guidelines: http://journals.plos.org/plosone/s/licenses-and-copyright.

Reviewers' comments:

Reviewer's Responses to Questions

**Comments to the Author**

1. Is the manuscript technically sound, and do the data support the conclusions?

Reviewer #1: Yes

Reviewer #2: Partly

Reviewer #3: Yes

Reviewer #4: Partly

2. Has the statistical analysis been performed appropriately and rigorously? 

Reviewer #1: Yes

Reviewer #2: No

Reviewer #3: Yes

Reviewer #4: Yes

3. Have the authors made all data underlying the findings in their manuscript fully available?

Reviewer #1: Yes

Reviewer #2: Yes

Reviewer #3: Yes

Reviewer #4: No

4. Is the manuscript presented in an intelligible fashion and written in standard English?

Reviewer #1: Yes

Reviewer #2: No

Reviewer #3: Yes

Reviewer #4: Yes

5. Review Comments to the Author

Reviewer #1: Dear authors

Thank you very much for this manuscript which is very well structured and written clearly. It adds one piece to the complex puzzle within the research topic of bird and bat collisions at wind turbines. I fully agree with the authors that it might be much more constructive for the protection of birds and bats to invest in technologies such as smart-curtailment approaches than just trying to specify time frames where wind turbines must be shut down.

Please find my questions and comments below. I suggest adding some of the information to your manuscript.

1. As far as I understood, the numbers of carcasses are not corrected for the detection probability (searcher efficiency together with persistence time and probability that a carcass is lying on the searched area). The detection probability might differ especially between ecoregions and for sure depending on carcass size. Do you have an estimate on how the carcass numbers would look like if the numbers were corrected for the detection probability? Maybe you have a few studies where these correction factors were included depending on ecoregion? Just to get an impression on the order of magnitude of the influence. Do you think the patterns you found could look differently if you had data to consider detection probability?

2, Do the carcass data all originate from wind farms where no kind of curtailment was implemented? If not, this would be a problem as this might cover seasonal patterns. Or it would show that the curtailment does not work because you still found the patterns.

3. Causes of collisions risk might also be related to characteristics of wind turbines (e.g. hub height and rotor size) or wind turbines sites (e.g. topographic characteristics). Did you check such kinds of factors in your models?

4. You had to exclude a lot of data because metadata of carcass searches were missing. Is it planned to conduct any actions to improve data quality delivered to AWWIC? Such data might be also relevant in future to check the efficiency of smart-curtailment approaches.

Reviewer #2: This paper covers a large spatial area and uses a large publicly available dataset of wind turbine collision surveys of birds and bats. This field of study is important to promote responsible green energy development. This paper has the potential to inform curtailment and collision mitigation efforts to support conservation on a broad scale.

The clarity of writing and language used makes the paper hard to understand in some places.

Examples:

lines 85-89

87-89 suggest changing to: “In addition, we compared studies with smaller datasets and spatial coverage to data that covered a much larger area.”

100-101 suggest changing to: “As such, it offers opportunities to study broad patterns across the United States.”

110-115

116-120 Explain further, unclear on the methods of exclusions

Some sentences are very unclear, and the placement of certain information presented does not fall into the proper paper sections (i.e., hypothesis/prediction in the methods; eg lines 138-143, 146-149).

Lines 131-132: Describe this exclusion further (# of species and observations removed). Maybe these exclusions would reveal important patterns for understudied species.

The language used (not stylistic preference) has obscured some of the key details of the paper and left some confusion (i.e., lines “422-227. Ultimately, the combination of stochasticity and non-stationarity in the ecological processes that underlie collision risk may render investments in technologies that reduce bat fatalities without requiring unnecessary curtailment of electricity generation, such as deterrents or smart-curtailment approaches, more effective than continued efforts to define risky periods a priori via fatality monitoring."

352-355

359-360

378-382

and other examples

Word choice makes it very difficult to extract scientific information. Writing can be improved and condensed significantly.

Further, some of the methods and data analysis with models is not fully presented and explained (i.e., model validation criteria, goodness of fit and other parameters), and the functions used in R should be further explained. The methods need additional details for clarity and reproducibility. E.g.,

Lines 197-199

200-209

214-219

Little reference to other key literature in the field to support some statements and reasons for methodology (e.g., lines 150,157-159).

Further discussion of the study’s limitations would improve the paper. Lines 399-405 discuss some, but these conclusions are not thoroughly explained and laid out.

Discussion of standardization on PCM can be expanded on and recommend specific improvements based on the use of such a large database.

Lastly, there is little comparison of results to other literature in the field. In particular, other research about collision patterns in the different eco-regions and taxa within smaller studies (i.e., does a small study scale up? How does this compare? and how does this study’s results compare to other research done on a similar scale like in Europe or Asia).

Reviewer #3: The authors looked at a large data set from bird and bat collisions with wind turbines across the United States. This study provides an interesting look at the seasonal differences in fatalities of bat species and bird guilds. Overall, I have very few concerns with the manuscript. Most of my comments below are minor edits or areas where a bit more information or clarification could be added. One small concern that I have is that the last sentence of the Abstract suggests that this study “highlights the value of pooling data…” but the Discussion focuses heavily on the inadequacies of the study. The authors might consider highlighting the values of the study a bit more in the Discussion (don't undersell the study!).

Reviewer #4: Reviewer Recommendations

Please provide:

Please provide a detection map. Some graphic example to visualize the search radius and data collected in any typical PCM.

Please provide:

Bar graph of detected bat species/ bird guilds- overall detections- worse hit species for bats/ and worse hit guilds for birds.

Line 45 Citation

Bose A, Dürr T, Klenke RA, Henle K (2020) Predicting strike susceptibility and collision patterns of the common buzzard at wind turbine structures in the federal state of Brandenburg, Germany. PLoS ONE 15(1): e0227698. doi:10.1371/journal.pone.0227698

Line 129 Citation

Bose, A., Dürr, T., Klenke, R.A. et al. Collision sensitive niche profile of the worst affected bird-groups at wind turbine structures in the Federal State of Brandenburg, Germany. Sci Rep 8, 3777 (2018). https://doi.org/10.1038/s41598-018-22178-z

Line 206 Citation

Bose, A., Dürr, T., Klenke, R.A. et al. Collision sensitive niche profile of the worst affected bird-groups at wind turbine structures in the Federal State of Brandenburg, Germany. Sci Rep 8, 3777 (2018). https://doi.org/10.1038/s41598-018-22178-z

Erickson, W., Wolfe, M., Bay, K., Johnson, D. & Gehring, J. L. A comprehensive analysis of small-passerine fatalities from collision with turbines at wind energy facilities. PLOS ONE. 9(9), e107491 (2014)

Line 309

Kindly change the line to “Based on data collected” and no means of verification of the statement is provided.

6. PLOS authors have the option to publish the peer review history of their article (what does this mean?). If published, this will include your full peer review and any attached files.

Reviewer #1: No

Reviewer #2: No

Reviewer #3: No

Reviewer #4: **Yes: **Dr. Anushika Bose

---

## [Author Response · Author response to Decision Letter 0]

14 Mar 2023

Comment: As far as I understood, the numbers of carcasses are not corrected for the detection probability (searcher efficiency together with persistence time and probability that a carcass is lying on the searched area). The detection probability might differ especially between ecoregions and for sure depending on carcass size. Do you have an estimate on how the carcass numbers would look like if the numbers were corrected for the detection probability? Maybe you have a few studies where these correction factors were included depending on ecoregion? Just to get an impression on the order of magnitude of the influence. Do you think the patterns you found could look differently if you had data to consider detection probability?

Our response: The reviewer is correct that we used raw carcass counts, uncorrected for detectability. We chose not to estimate detectability because it would not affect the patterns that we observed. Detectability during PCM is not estimated in a time-specific fashion but rather is calculated at the level of the study, which may cover multiple years of collision records. As such, although incorporating detectability would change the estimated number of carcasses present, it would have no effect on the temporal pattern of carcass recoveries. For example, imagine a PCM that yielded a detectability estimate of 0.30: applying this to a study would produce a 30% inflation in the estimated number of collisions, but would inflate observed carcass counts equally across the year, thus resulting in no change to the temporal pattern of collision counts. In this manuscript, we do not discuss or interpret the absolute number of carcasses observed (whether corrected for detectability or not) and thus incorporating detectability estimates will yield no change in our reported results or conclusions. 

The same reasoning applies to the reviewer’s concern regarding variation in detectability among ecoregions. Although we are not aware of any studies showing that such variation exists, even if it did ecoregional variation in detectability would not change our results or conclusions. Higher or lower detectability in one ecoregion versus another might change the magnitude of any differences in observed carcass counts, but would not change the relative patterns of temporal variation. That is, differences in the timing of peak carcass counts between ecoregions are independent of any differences in detectability; an earlier (or later) peak in one ecoregion will always be earlier (or later) regardless of whether detectability varies between those ecoregions. 

Comment: Do the carcass data all originate from wind farms where no kind of curtailment was implemented? If not, this would be a problem as this might cover seasonal patterns. Or it would show that the curtailment does not work because you still found the patterns.

Our response: We thank the reviewer for raising this question, as it is an important issue to discuss and we have added text to clarify that the data used in this study include sites with and without curtailment (see below for our proposed changes). Regions with bat species of concern (Threatened, Endangered, or otherwise) have especially high rates of curtailment in the studies to which we had access. Regions without these species have fewer sites with curtailment. Nonetheless, we do not believe that the presence or absence of curtailment will substantively alter our results or conclusions. 

First, curtailment is nearly always done in response to concerns about bat deaths, and thus occurs at night. Because of the diurnal habits of most birds, and because nocturnal migrants almost always fly much higher than the rotor-swept area, curtailment is not expected to benefit birds; this assumption is supported by empirical evidence (see DOI: 10.1002/jwmg.21844). Informed, or smart, curtailment can reduce deaths of soaring birds, but none of the studies in this analysis relied on this new and not-yet-widely-adopted technology. As such, we conclude that curtailment is unlikely to influence the seasonal patterns of bird collisions that we describe.

Curtailment is generally effective in reducing bat mortality from collisions with turbines, and thus could conceivably influence our results regarding bats. However, we consider it unlikely that variation in curtailment regimes biased our results in any substantive way. Most importantly, curtailment, when required, is typically imposed only during peak periods of bat activity, usually late summer and fall. As such, curtailment would tend to dampen seasonal peaks in collision fatalities, while having no effect on collision counts during the rest of the year, rendering our analysis conservative in terms of identifying periods of highest risk. Curtailment would tend to diminish our ability to detect seasonal peaks but we cannot conceive of any realistic situation or temporal distribution of collisions in which curtailment would artificially introduce spurious peaks. 

Similarly, even though the extent of curtailment seems to vary on a region-by-region basis, we don’t believe that this introduces bias that alters our conclusions. The greatest concern in regards to regional variation in curtailment would be that it affects among-region comparisons of timing. As our analysis does not evaluate differences in the absolute magnitude of carcass counts, any dampening effect of curtailment would not influence our findings. Further, because curtailment is generally applied during the same window of time in all areas (July – October; see https://doi.org/10.1002/we.2741), we do not expect a region-by-curtailment date interaction that would affect the shape of the seasonal curves in carcass counts. The most likely influence of regional variation in the frequency of curtailment would be to depress peaks in regions with curtailment while leaving the timing of peak carcass counts unaffected in regions without curtailment. Although this might lead us to conclude that one region showed a peak in fatalities whereas another region did not, regional variation in curtailment would not lead us to draw false conclusions about the timing of peaks. Importantly, we do not see this pattern with bats, as all ecoregions showed significant peaks in bat fatalities (which was not the case with birds). 

From a practical standpoint, incorporating curtailment as a covariate is difficult, especially in a large-scale analysis such as ours. Curtailment poses a challenge for a multi-study analysis because it is imposed in a site-specific fashion that is not always described in the pooled data in AWWIC. For example, some sites identified as incorporating curtailment may in fact only impose curtailment on one or a few turbines, generally those proximal to some sensitive resource, like hibernacula, leaving the rest of the turbines uncurtailed. A PCM conducted at this site would thus include a mix of curtailed and uncurtailed turbines, with no way to distinguish which carcasses came from which turbines without more information than we had access to. Other sites identified as curtailed may impose blanket curtailment on every turbine. As such, even though we can distinguish curtailed and uncurtailed projects, we cannot differentiate among the different forms of curtailment and would thus still be left comparing projects that had employed very different regimes. The only sure solution for excluding any potential effect of curtailment would be to exclude any project that reported any form of curtailment, which would leave very few PCM for analysis, as most projects – especially those from recent years – include at least occasional curtailment. As our sample size is already small, the result would almost surely be a complete or near-complete loss of information on seasonal patterns of collision fatalities. Given that our results show peaks in bat fatalities even in regions where curtailment is widespread, and given the theoretical reasons to expect that curtailment will not bias estimates of seasonality in peak fatality periods (as described above), we do not believe that this strategy would improve the strength of inference. Instead, it would effectively serve to mask all of the potentially important patterns we describe. 

In conclusion, we agree with the reviewer that these points should be discussed in the manuscript, but we do not believe that our decision to include curtailed and uncurtailed projects in our analysis is likely to change our conclusions. Furthermore, we do not believe that it would be possible to analytically address this issue with the data to which we have access, especially as we have sought to include as extensive a data set as possible, which necessarily involves accepting some measure of among-study variation in unmeasured variables. 

In summary, to address this concern, we added the following text to the discussion:

“Data used in this analysis were drawn from PCM conducted at facilities with variable curtailment regimes. Some imposed curtailment, whereas others did not. We do not believe that variation in curtailment regime was likely to have introduced substantial bias into our results nor do we believe that it influenced our conclusions in any significant fashion. First, any effect of curtailment should be negligible for birds. In the U.S., curtailment is typically imposed during the evening to reduce bat fatalities. The largely diurnal habits of the birds considered in this analysis and the tendency of nocturnal migrants to fly well above the rotor-swept area should mean that curtailment as currently practiced will have negligible effects on bird fatalities. Some empirical evidence supports this assumption [25].

Curtailment is generally effective in reducing bat fatalities, however [26], and thus may have affected the number of carcasses recovered during PCM. Nonetheless, we think that variation in curtailment regimes is unlikely to affect our results or conclusions. Curtailment in the U.S. is usually imposed during a specific window of time when bat fatalities are expected to be greatest, typically July – October [24]. If effective, curtailment should depress seasonal peaks in fatality counts but will not introduce spurious peaks in carcass counts. This means that curtailment would most likely render our tests for significant seasonal peaks conservative, making it more difficult to discern statistically significant peaks in carcass counts. It should not, however, create artificial patterns of seasonality.”

Comment: You had to exclude a lot of data because metadata of carcass searches were missing. Is it planned to conduct any actions to improve data quality delivered to AWWIC? Such data might be also relevant in future to check the efficiency of smart-curtailment approaches.

Our response: Yes, changes will be made to improve the usefulness of data submitted to AWWIC. In particular, as we write in the manuscript, “Going forward, all new studies contributed to AWWIC will be required to include search schedules and exact dates of carcass discovery, which will ensure that future efforts to model collision timing will not be restricted for similar reasons.”

Comment: 87-89 suggest changing to: “In addition, we compared studies with smaller datasets and spatial coverage to data that covered a much larger area.”

Our response: text changed accordingly. 

Comment: 100-101 suggest changing to: “As such, it offers opportunities to study broad patterns across the United States.”

Our response: text changed accordingly.

Comment: 110-115: The clarity of writing and language used makes the paper hard to understand in some places.

Our response: we have added text “(e.g., different time intervals between searches)” to clarify that the problem we sought to avoid was that different PCM efforts may have different intervals of time between searches, which could lead to inter-study variation in the accuracy of estimated dates of collision fatalities.

Comment: 116-120 Explain further, unclear on the methods of exclusions.

Our response: we have added text “all of our models (see below) included an offset term for the number of searches conducted per day, an effort to account for variable levels of effort on each search day, and thus incidental carcass discoveries could not be included” to explain why incidental discoveries could not be accommodated in our modelling approach.

Comment: Some sentences are very unclear, and the placement of certain information presented does not fall into the proper paper sections (i.e., hypothesis/prediction in the methods; eg lines 138-143, 146-149).

Our response: We agree that the placement of these statements may be confusing. To address this critique, we have removed the reference to expected patterns of variation among guilds from the Methods, as these patterns are discussed in greater detail in the Discussion. 

Comment: Lines 131-132: Describe this exclusion further (# of species and observations removed). Maybe these exclusions would reveal important patterns for understudied species.

Our response: The information requested is presented by reference to [12], which includes total numbers of observations by species, including those excluded to sample-size considerations from the present analysis.

Comment: The language used (not stylistic preference) has obscured some of the key details of the paper and left some confusion (i.e., lines “422-227. Ultimately, the combination of stochasticity and non-stationarity in the ecological processes that underlie collision risk may render investments in technologies that reduce bat fatalities without requiring unnecessary curtailment of electricity generation, such as deterrents or smart-curtailment approaches, more effective than continued efforts to define risky periods a priori via fatality monitoring."

Our response: We have changed the text in an effort to clarify our point that the analysis of PCM data may not be the most efficient path forward: “Spatial and temporal variability in observed patterns of collision timing, along with the potential that the phenology of affected species is undergoing directional change due to a warming climate, means that it may be difficult to accurately predict the optimal time to curtail turbines, even with more intensive PCM and expansion of the spatial coverage of AWWIC. If accurate prediction of when collision fatalities are most likely to occur remains unachievable, then investments in technologies that reduce bat fatalities without requiring unnecessary curtailment of electricity generation, such as deterrents or smart-curtailment approaches, may prove more effective than continued efforts to define risky periods a priori via fatality monitoring.”

Comment: Further, some of the methods and data analysis with models is not fully presented and explained (i.e., model validation criteria, goodness of fit and other parameters), and the functions used in R should be further explained. The methods need additional details for clarity and reproducibility. E.g.,

Lines 197-199

200-209

214-219

Our response: We have attempted to clarify the referenced lines of text. 

Comment: Further discussion of the study’s limitations would improve the paper. Lines 399-405 discuss some, but these conclusions are not thoroughly explained and laid out.

Our response: Thank you for this suggestion. We have revised the discussion to more thoroughly address limitations of the study.

Comment: Discussion of standardization on PCM can be expanded on and recommend specific improvements based on the use of such a large database.

Our response: We have included additional information and references on PCM standardization.

Comment: there is little comparison of results to other literature in the field. In particular, other research about collision patterns in the different eco-regions and taxa within smaller studies (i.e., does a small study scale up? How does this compare? and how does this study’s results compare to other research done on a similar scale like in Europe or Asia).

Our response: The first two questions are addressed in the discussion, in particular “We conclude that most of what is known about seasonal variation in collisions among birds and bats at wind facilities in the U.S. holds true even as sample size increases by an order of magnitude,”. We do not believe that the third question, how our results compare to similarly scaled studies in Europe or Asia, can be answered at present as we found no similar studies from any other region during our literature review. Indeed, to our knowledge, AWWIC is unique in providing pooled data on collision fatalities at such a broad spatial scale.

Comment: One small concern that I have is that the last sentence of the Abstract suggests that this study “highlights the value of pooling data…” but the Discussion focuses heavily on the inadequacies of the study. The authors might consider highlighting the values of the study a bit more in the Discussion (don't undersell the study!).

Our response: Thank you for pointing out this inconsistency. We agree that the manuscript did not clearly establish why pooling data might be useful. To address this concern, we made the following changes to the text. First, we added a phrase to the opening of the Discussion that makes clear that some of the findings we report were possible only because of the pooled data. Next, we added the following sentence to the second paragraph of the Discussion:

“Pooling PCM data in AWWIC proved especially useful in allowing a finer-grained analysis of bird collisions. In particular, we found distinct patterns in the seasonal timing of collision fatalities among bird guilds, which has not been described previously.”

We go on to explain that this finding suggests a new hypothesis explaining why collision fatalities vary among bird species and we outline how the hypothesis might be tested (“This hypothesis could be tested by examining the relationship between collision fatalities and the amount of time different species spend in the vicinity of operating turbines, for example by combining data on the location of turbines in the U.S. [15], collision fatalities in AWWIC, and intra-annual distribution models generated by eBird (eBird Status and Trends products; ebird.org/science/status-and-trends).”). 

We have also added text to highlight the novelty of our findings for Mexican free-tailed bats, a species limited to the southern parts of the United States for which the seasonal timing of collisions has not been widely reported:

“However, the geographically extensive data in AWWIC allowed us to compare for the first time the timing of fatalities for these more widespread species with Mexican free-tailed bats”

Finally, we added text to clarify that the manuscript offered new insight into how temporal patterns in the number of collisions varied across latitudes, which may also help inform region-specific curtailment prescriptions:

“Using data from a large geographic area also allowed us to examine how temporal patterns in the number of collisions varied across latitudes, a question that cannot be addressed by site- or region-specific studies. Among the migratory tree bats, species that occurred across multiple ecoregions tended to show earlier fatality peaks in more northerly ecoregions. For example, when we compared the estimated peak date of fatalities for Hoary Bat between adjacent ecoregions occupying largely non-overlapping latitudinal ranges, the more northerly ecoregion always showed an earlier peak date. The peak date of Hoary Bat fatalities in the West-Central Semiarid Prairies was 17 August versus 4 September in the South Central Semiarid Prairies; 12 August in the Temperate Prairies versus 23 August in the Ozark/Ouachita-Appalachian Forests; and 10 August in the Mixed Wood Plains versus 23 August in the Ozark/Ouachita-Appalachian Forests. The same patterns were evident for both Eastern Red Bat and Silver-haired Bat.” 

And:

“Analyzing data from across the U.S. also offered potentially useful new insight into the potential to apply region-specific curtailment practices.”. 

Comment: Line 45 Citation

Bose A, Dürr T, Klenke RA, Henle K (2020) Predicting strike susceptibility and collision patterns of the common buzzard at wind turbine structures in the federal state of Brandenburg, Germany. PLoS ONE 15(1): e0227698. doi:10.1371/journal.pone.0227698

Our response: Citation added. 

Comment: Line 129 Citation

Bose, A., Dürr, T., Klenke, R.A. et al. Collision sensitive niche profile of the worst affected bird-groups at wind turbine structures in the Federal State of Brandenburg, Germany. Sci Rep 8, 3777 (2018). https://doi.org/10.1038/s41598-018-22178-z

Our response: Citation added.

Comment: Line 206 Citation

Bose, A., Dürr, T., Klenke, R.A. et al. Collision sensitive niche profile of the worst affected bird-groups at wind turbine structures in the Federal State of Brandenburg, Germany. Sci Rep 8, 3777 (2018). https://doi.org/10.1038/s41598-018-22178-z

Erickson, W., Wolfe, M., Bay, K., Johnson, D. & Gehring, J. L. A comprehensive analysis of small-passerine fatalities from collision with turbines at wind energy facilities. PLOS ONE. 9(9), e107491 (2014)

Our response: Citations added. 

Comment: Line 309

Kindly change the line to “Based on data collected” and no means of verification of the statement is provided.

Our response: Text changed as requested. 

Comment: Lines 38 to 39 – How did the authors determine which birds belonged to which guilds? The authors might include adding reference(s).

Our response: Text and citation added.

Comment: Line 56 - Why did the spring peak exceed the fall in magnitude? Could the authors add a few more words to explain?

Our response: We could not find a reference to spring or fall peaks on line 56.

Comment: Line 60 – Similar comment to previous. Why do bats not show a spring peak in fatalities?

Our response: This statement refers to results from previous studies, none of which provide an answer as to why migratory bats do not show spring peaks in fatalities. At present, it is our understanding that the answer to this question remains unknown.

Comment: Line 182 – Remove “models” after “GAMM” to read “We used GAMMs first to describe...”

Our response: Text changed accordingly.

Comment: Lines 292 to 293 – “Most of the improvement in model performance seems to be related to profound differences in seasonal timing of fatalities among guilds.” Perhaps I’m misunderstanding but this sentence reads awkwardly to me. Maybe change to “Most of the variation in model performance...” rather than “improvement” or change the second half of the sentence. It’s the “improvement related to differences” that doesn’t make sense to me.

Our response: Thank you for the suggested change. We have modified the text to read “Most of the improvement in model performance seems to have occurred because the interaction of guild and ecoregion helped to account for the observed differences in seasonal timing of fatalities among guilds (Fig 5)” so as to explain that we think that the improved model fit is related to the parameter allowing for an interaction between guild and ecoregion, which helps to account for variation associated with differences in fatality timing among guilds and across ecoregions.

Comment: “Lines 334 to 335 – Species that occurred across multiple ecoregions tended to show earlier fatality peaks in more northerly ecoregions.” – Why? Could the authors add a few words or a sentence explaining this?

Our response: The text cited in this comment has been moved around a bit, but we have added the following statement to clarify why we believe that fatalities peak earlier further north: 

“We assume that this latitudinal gradient in the timing of maximum fatality counts reflects the gradual southward movement of a broad migratory front of bats, perhaps triggered by favorable weather conditions that encouraged large numbers of individuals to initiate large-scale movements [4,9,23].”

Comment: Line 357 – Why was year a factor that complicated results into practice? Explain.

Our response: We have attempted to clarify that year-to-year and among-site variation makes model predictions of fatality peaks in future years at any given site difficult: “In other words, our best-supported model indicated that the seasonal timing of fatalities varied from year-to-year and among sites, which limits our ability to predict future peaks in fatalities at any given site.”

Comment: Lines 359 to 360 – “Absent robust, site-specific data, longer curtailment periods will be required to account for uncertainties in when fatalities will peak at any given site in any given year.” This reads awkwardly. Maybe change to “In the absence of robust, site-specific data...”?

Our reponse: Based on different comments from another reviewer about this statement, we changed the text to read:

“Absent ongoing PCM at many locations across the U.S…”, which hopefully also addresses the concern about the awkwardness of the previous phrasing. 

Comment: Line 370 to 371 – “Going forward, all new studies contributed to AWWIC will include search schedules and exact dates of carcass discovery...” Why is this? Is this a new requirement of AWWIC? Might be worth stating this.

Our response: We have added text to clarify that this will be a requirement for future submissions to AWWIC.

Comment: Line 398 – Remove one of these words “including specifying”.

Our response: We have deleted “specifying”.

Comment: Lines 418 to 420 – “That climate change is inducing directional changes in the timing of key events in the life cycle of many organisms, including potentially in the timing of bird and bat collisions with turbines, is more difficult to address.” Awkward wording. Maybe switch this around so it reads “It is more difficult to address if climate change is...” This point also seems a bit out of place. A reference to climate change seems important but a bit out of nowhere. I suggest expanding or removing.

Our response: Thank you for this suggestion. We agree that the point is not made clearly and also agree that some of the themes raised in these paragraphs were not adequately developed. We have deleted these two paragraphs as they were both confusing to readers and ultimately seemed unnecessary.

Comment: Figure 1 – Maybe it’s the quality of the figure in this version but some of the text is a bit difficult to read. Particularly the 9.2 label over the dark blue region. Perhaps this one could be switched to white text?

Our response: The labels appear legible in the high-resolution versions that we reviewed. If the problem persists even when viewing the high-resolution images, then we can adjust the font color for the problematic text. 

Comment: Figure 4 – Again, this is likely a function of low-quality images, but the colours and the boundaries of the shaded areas are difficult to see. Ignore this comment if the version that will be published is clear.

Our response: The labels appear legible in the high-resolution versions that we reviewed. If the problem persists even when viewing the high-resolution images, then we can adjust the font color for the problematic text. 

Comment: Please provide a detection map. Some graphic example to visualise the search radius and data collected in any typical PCM.

Our response: An excellent figure and non-technical description of a typical PCM was provided in a recent review paper, and we have added text to direct interested readers to this reference:

“Requirements for PCM vary by state and Federal permitting jurisdiction but have been a common practice since the Land-based Wind Energy Guidelines were published [11] (for an illustration and description of what constitutes a typical PCM, see [1])”.

Comment:

Please provide: Bar graph of detected bat species/ bird guilds- overall detections- worse hit species for bats/ and worse hit guilds for birds.

Our response: Complete data on the number of collisions per species are available in [12] and [14]. We have added text to clarify this:

“The included studies accounted for 14,721 search days and 312,510 total searches. In total, 10,291 bat carcasses and 3,789 bird carcasses with exact find dates were available for this analysis (S2 and S3 Tables; for total carcasses by species considered in this analysis, see [12,14])”

---

## [Editor Report · Decision Letter 1]

10 Apr 2023

Seasonal patterns of bird and bat collision fatalities at wind turbines

PONE-D-22-30951R1

Dear Dr. Lloyd,

We’re pleased to inform you that your manuscript has been judged scientifically suitable for publication and will be formally accepted for publication once it meets all outstanding technical requirements.

Kind regards,

William David Halliday, Ph.D.

Academic Editor

PLOS ONE

Additional Editor Comments (optional):

Very good job on your revisions. The study was well executed, and the paper well written.
---

## [Editor Report · Acceptance letter]

12 Apr 2023

PONE-D-22-30951R1 

Seasonal patterns of bird and bat collision fatalities at wind turbines 

Dear Dr. Lloyd:

I'm pleased to inform you that your manuscript has been deemed suitable for publication in PLOS ONE. Congratulations! Your manuscript is now with our production department. 

Kind regards, 

on behalf of

Dr. William David Halliday 

Academic Editor

PLOS ONE